# STABLE SIGNATURE IS UNSTABLE: REMOVING IMAGE WATERMARK FROM DIFFUSION MODELS

## ABSTRACT

Watermark has been widely deployed by industry to detect AI-generated images. A recent watermarking framework called *Stable Signature* (proposed by Meta) roots watermark into the parameters of a diffusion model's decoder such that its generated images are inherently watermarked. Stable Signature makes it possible to watermark images generated by *open-source* diffusion models and was claimed to be robust against removal attacks. In this work, we propose a new attack to remove the watermark from a diffusion model by fine-tuning it. Our results show that our attack can effectively remove the watermark from a diffusion model such that its generated images are non-watermarked, while maintaining the visual quality of the generated images. Our results highlight that Stable Signature is not as stable as previously thought.

## 1 INTRODUCTION

With the rapid development of generative AI (GenAI), it becomes increasingly more difficult to distinguish AI-generated and non-AI-generated images. The misuse of AI-generated images presents a significant risk of spreading misinformation. Watermarking (Bi et al., 2007; Zhu et al., 2018; Zhang et al., 2020; Tancik et al., 2020; Fernandez et al., 2023; Wen et al., 2023; Jiang et al., 2024) has emerged as a crucial technology for detecting AI-generated images and been widely deployed by industry. For instance, OpenAI incorporates a watermark into images generated by DALL-E (Ramesh et al., 2021); Stability AI deploys a watermarking technique in Stable Diffusion (Rombach, 2022); and Google has introduced SynthID as a watermarking solution for images generated by Imagen (Saharia et al., 2022). In watermark-based detection, a watermark is embedded in AI-generated images before they are accessed by users. During detection, if the same watermark can be extracted from an image, it is identified as AI-generated.

Image watermark can be categorized into three groups based on the timing when watermark is embedded into AI-generated images: *post-generation*, *pre-generation*, and *in-generation*. Post-generation watermark (Luo et al., 2020; Bi et al., 2007; Zhu et al., 2018; Zhang et al., 2020; Al-Haj, 2007; Tancik et al., 2020; Jiang et al., 2024) embeds a watermark into an image after the image has been generated, while pre-generation watermark (Wen et al., 2023) embeds a watermark into the initial noisy latent vector of a diffusion model. However, these watermarking methods are vulnerable when the diffusion models are open-source. In particular, an attacker can easily remove the watermarking components from the open-source diffusion model to generate non-watermarked images. In contrast, in-generation watermark (e.g., Stable Signature (Fernandez et al., 2023) and WOUAF (Kim et al., 2024)) roots watermark directly into the parameters of a diffusion model's decoder. It enables the images generated by the diffusion model to be inherently watermarked without introducing any external watermarking components. This method is particularly suited for watermarking images generated by open-source diffusion models.

Watermark removal attacks aim to remove watermarks from watermarked images, and can be divided into two types: *per-image-based* and *model-targeted*. Per-image-based attacks (Jiang et al., 2023; An et al., 2024; Lukas et al., 2024; Zhao et al., 2023; Saberi et al., 2024) add a carefully crafted perturbation to each watermarked image individually. These removal attacks need to process watermarked images one by one, which is inefficient when removing watermarks from a large volume of watermarked images. In contrast, model-targeted attacks directly modify a diffusion model's parameters to make its generated images non-watermarked. For instance, Fernandez et al.

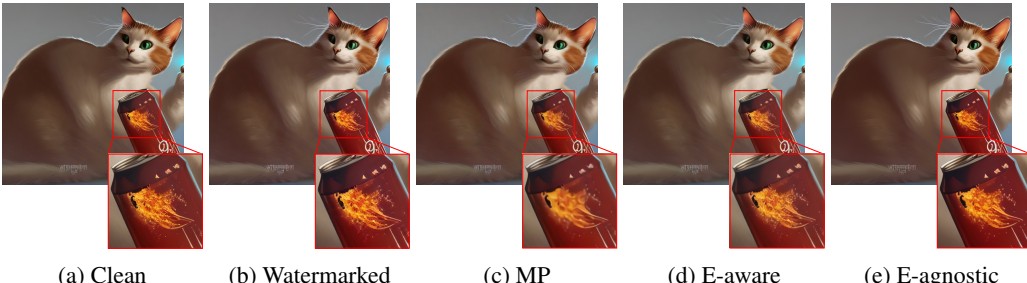

| (a) Clean | (b) Watermarked | (c) MP | (d) E-aware | (e) E-agnostic |

Figure 1: An example of image generated by (a) the clean Stable Diffusion 2.1, (b) Stable Diffusion 2.1 watermarked by Stable Signature, (c) watermarked Stable Diffusion 2.1 fine-tuned by MP, (d) watermarked Stable Diffusion 2.1 fine-tuned by our attack with access to the encoder, and (e) watermarked Stable Diffusion 2.1 fine-tuned by our attack without access to the encoder. The same denoised latent vector is used by all diffusion models' decoders to generate the images. The watermark can only be detected in the image generated by (b). The image generated by (c) has significant loss of details.

(2023) also proposed a model-targeted removal attack, called *model purification (MP)*, to attack Stable Signature. However, MP requires access to the diffusion model's encoder, and the model provider can easily defend against this by making the encoder closed-source, as it is not necessary for image generation. Moreover, MP significantly deteriorates image quality (Fernandez et al., 2023; Kim et al., 2024), based on which Stable Signature and WOUAF were claimed to be robust against model-targeted removal attacks.

In this work, we propose a new model-targeted attack to remove in-generation watermark from open-source diffusion models. Our attack fine-tunes a diffusion model's decoder using a set of non-watermarked images, which we call *attacking dataset*. Specifically, our attack consists of two steps. In Step I, we propose different methods to estimate a *denoised latent vector* for each non-watermarked image in the attacking dataset in two settings, i.e., with and without access to the diffusion model's encoder. The open-source diffusion model's decoder takes a denoised latent vector as input and outputs a watermarked image that is visually similar to the corresponding non-watermarked image. In Step II, we leverage the non-watermarked images in the attacking dataset and their corresponding estimated denoised latent vectors to fine-tune the diffusion model's decoder to remove the watermark from it. Our key idea is to fine-tune the decoder such that its generated images based on the denoised latent vectors are close to the corresponding non-watermarked images in the attacking dataset.

We empirically evaluate our attack on the open-source diffusion models, i.e., Stable Diffusion 2.1 which is watermarked by Stable Signature and Stable Diffusion 2-base which is watermarked by WOUAF. Our results show that our attack can effectively remove the watermark from the diffusion models such that their generated images are non-watermarked, while maintaining image quality. Moreover, our attack substantially outperforms MP, the only existing model-targeted removal attack (Fernandez et al., 2023), in the scenario in which it is applicable. As shown in Figure 1, our attack can retain most information in the image after removing the watermark, while MP results in a blurry image with significant loss of details. Our results suggest that Stable Signature is not as robust as previously thought, and the design of a robust watermarking strategy for images generated by open-source diffusion models remains an open challenge.

## 2 RELATED WORKS

### 2.1 LATENT DIFFUSION MODEL

Diffusion models (Dhariwal & Nichol, 2021; Ho et al., 2020; Kingma et al., 2021; Ho et al., 2022) exhibit exceptional capability in generating images. A *latent diffusion model* (Rombach et al., 2022) performs the diffusion process in the latent space, enhancing efficiency in both training of the diffusion model and image generation. A latent diffusion model has four main components: an encoder $E$ to encode an image $x$ into a *latent vector* $E(x)$, diffusion process $DP$ to add Gaussian noise to the latent vector to obtain a *noisy latent vector* $z_T = DP(E(x))$ where $T$ denotes the number of steps

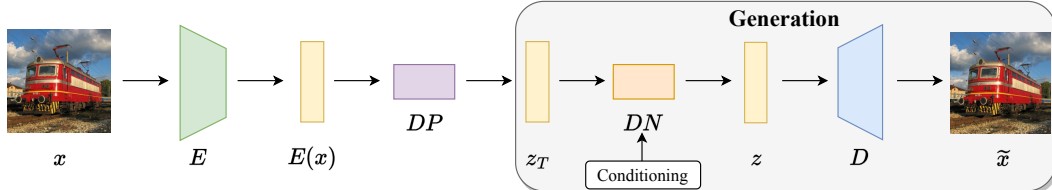

Figure 2: The main components of a latent diffusion model.

in diffusion process, denoising layers $DN$ to obtain a *denoised latent vector* $z = DN(z_T, c)$ where $c$ denotes the conditioning such as a text prompt or a depth map, and a decoder $D$ to reconstruct an image $D(z)$ from $z$. The diffusion process is a predefined probabilistic process that iteratively adds Gaussian noise to a latent vector, while the remaining three components are learnt using an image dataset. During image generation, a noisy latent vector $z'_T$ is sampled from Gaussian distribution, and the denoising layers $DN$ and decoder $D$ are used to generate an image $D(DN(z'_T, c))$. The main components of a latent diffusion model are shown in Figure 2.

## 2.2 IMAGE WATERMARK

**Post-generation watermark:** Post-generation watermarking methods (Bi et al., 2007; Al-Haj, 2007; Zhu et al., 2018; Tancik et al., 2020; Wang, 2021; Luo et al., 2020; Jing et al., 2021; Jiang et al., 2024) embed watermarks into images after the image generation process. These methods typically consist of three main components: a watermark (represented as a bitstring), a watermarking encoder for embedding the watermark into an image, and a watermarking decoder for extracting the watermark from an image. These methods can be categorized into two groups based on how the encoder and decoder are designed: *learning-based* and *non-learning-based*. Learning-based methods (Zhu et al., 2018; Zhang et al., 2020; Tancik et al., 2020; Luo et al., 2020; Jiang et al., 2024) leverage deep learning techniques, utilizing neural networks for both encoding and decoding, while non-learning-based methods (Pereira & Pun, 2000; Al-Haj, 2007; Bi et al., 2007; Wang, 2021) rely on manually crafted encoding and decoding algorithms. In closed-source setting, where the diffusion model is proprietary and users can only interact with it through API, learning-based watermarking methods exhibit significant robustness against various attacks (An et al., 2024; Tancik et al., 2020; Jiang et al., 2023). In open-source setting, however, such robustness is compromised. An attacker can easily remove the watermarking components from the open-source diffusion model, thus generating non-watermarked images without constraints.

**Pre-generation watermark:** Pre-generation watermarking methods (Wen et al., 2023) embed watermark into images before the image generation process. In diffusion models, for instance, a watermark can be incorporated into the noisy latent vector $z_T$ (Wen et al., 2023). Subsequently, the image generated from this watermarked noisy latent vector contains the watermark. The watermark retrieval process involves an inverse operation of DDIM sampling (Song & Ermon, 2020), which reconstructs the noisy latent vector from the generated image. However, such pre-generation watermark is also vulnerable in open-source setting. An attacker can substitute the watermarked noisy latent vector with a non-watermarked one, which is drawn from a Gaussian distribution. Consequently, image generated from this non-watermarked noisy latent vector does not contain the watermark.

**In-generation watermark:** In-generation watermarking methods (Fernandez et al., 2023; Kim et al., 2024) modify the parameters of the diffusion model's decoder to ensure that all images generated by the model inherently contain a watermark. These methods seamlessly integrate the watermarking process into image generation. For example, Stable Signature (Fernandez et al., 2023) fine-tunes the diffusion model's decoder using the HiDDeN (Zhu et al., 2018) watermarking decoder. Once fine-tuned, each generated image embeds a predetermined watermark, which can be decoded by the watermarking decoder, effectively embedding the watermark within the model's parameters. Similarly, WOUAF (Kim et al., 2024) employs a trained mapping network and weight modulation technique to modify the diffusion model's decoder, instead of fine-tuning. These approaches are well-suited for open-source diffusion models, as they prevent attackers from easily removing the watermark by simply discarding the watermarking components.

## 2.3 WATERMARK REMOVAL ATTACKS

**Per-image-based:** Per-image-based removal attacks (Jiang et al., 2023; An et al., 2024; Lukas et al., 2024; Zhao et al., 2023; Saberi et al., 2024) involve adding a carefully crafted perturbation on each watermarked image to remove the watermark. Common image processing techniques, such as JPEG compression and contrast adjustment, can introduce a perturbation for the watermarked image to remove the watermark. Furthermore, more sophisticated per-image-based removal attacks can be employed if the attacker has access to the watermarking decoder or detection API. For instance, Jiang et al. (2023) proposed a white-box attack that assumes the attacker has access to the watermarking decoder, and a black-box attack that strategically manipulates the watermarked image based on detection API query results to remove the watermark. These per-image-based removal attacks are applicable to all three groups of watermarks mentioned above as they do not require access to the image generation process. However, they are inefficient when applied to a large volume of images due to the individualized design of perturbations for each watermarked image.

**Model-targeted:** Model-targeted removal attacks (Fernandez et al., 2023) are specifically designed for removing in-generation watermark. Such attacks involve modifying the diffusion model's parameters such that its generated images are non-watermarked. For instance, Fernandez et al. (2023) proposed MP to attack their Stable Signature in-generation watermark. This method aims to purify the diffusion model's decoder using non-watermarked images. However, it encounters challenges in effectively removing the watermark without significantly degrading image quality. Model-targeted removal attacks show high efficiency in removing watermark from numerous watermarked images, as it only requires a one-time modification of the diffusion model and images generated by the modified diffusion model are non-watermarked. These methods offer much higher efficiency compared to per-image-based removal attacks when handling numerous watermarked images.

## 3 PROBLEM FORMULATION

### 3.1 WATERMARKED DIFFUSION MODEL DECODER $D_w$

We denote by $D_c$ a clean diffusion model decoder without watermark. $D_c$ is fine-tuned as a watermarked diffusion model decoder $D_w$ such that its generated images are inherently embedded with a ground-truth watermark $w_g$. Formally, any generated image $D_w(DN(z_T, c))$ is embedded with $w_g$, where $z_T$ is a noisy latent vector sampled from a Gaussian distribution, $DN$ is the denoising layers, and $c$ is the conditioning. $D_w$ is made open-source, allowing users to generate watermarked images.

### 3.2 THREAT MODEL

**Attacker's goals:** Given a watermarked diffusion model decoder $D_w$, an attacker aims to fine-tune it as a non-watermarked diffusion model decoder $D_{nw}$. Specifically, the attacker aims to achieve two goals: 1) *effectiveness goal*, and 2) *utility goal*. The effectiveness goal means that images generated by $D_{nw}$ do not have the watermark $w_g$ embedded; while the utility goal means that the images generated by $D_{nw}$ maintain visual quality, compared to those generated by $D_w$.

**Attacker's knowledge:** A watermarked latent diffusion model consists of an encoder $E$, diffusion process $DP$, denoising layers $DN$, and a watermarked decoder $D_w$. The denoising layers $DN$ and decoder $D_w$ are involved when generating images, i.e., $D_w(DN(z_T, c))$ is a generated image, where $z_T$ is a noisy latent vector sampled from Gaussian distribution and $c$ is the conditioning. We assume $DN$ and $D_w$ are open-source, and thus the attacker has access to them. Depending on whether $E$ and $DP$ are open-source, we consider the following two scenarios:

- **Encoder-aware (E-aware).** In this scenario, the model provider also makes $E$ and $DP$ open-source. Therefore, the attacker has access to them. For instance, Stable Diffusion model makes its $E$ and $DP$ open-source.

- **Encoder-agnostic (E-agnostic).** In this scenario, $E$ and $DP$ are not open-source, e.g., because image generation only requires $DN$ and $D_w$. Therefore, the attacker does not have access to $E$ and $DP$ in this setting.

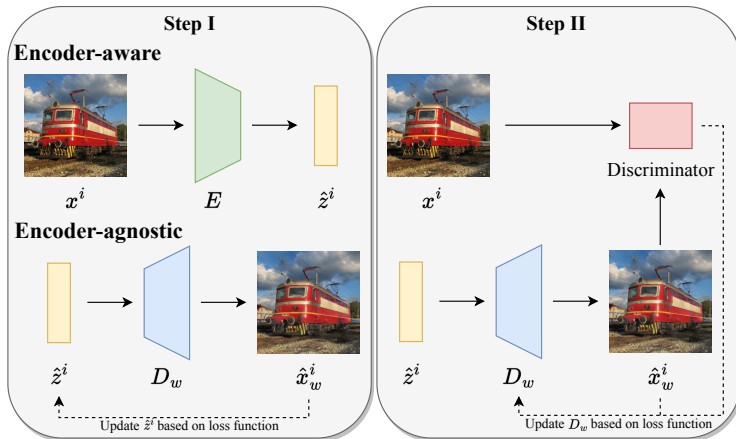

Figure 3: Overview of our attack. The solid arrows represent the direction of data flow and the dashed arrows represent the direction of gradient flow.

Additionally, we assume the attacker has access to a set of non-watermarked images, which we call attacking dataset. For instance, the attacker can simply use popular benchmark images (e.g., ImageNet) as the attacking dataset. The attacking dataset is used to remove watermark from the watermarked diffusion model decoder $D_w$.

**Attacker's capability:** We assume the attacker can modify the parameters of the open-sourced watermarked latent diffusion model decoder $D_w$. The denoising layers $DN$, which are much larger than the decoder, requires much more computational resources to modify. For instance, in Stable Diffusion 2.1, the denoising layers have about 10 times more parameters than the decoder. Therefore, we assume the attacker modifies the decoder.

## 4 OUR ATTACK

### 4.1 OVERVIEW

We propose a two-step method to fine-tune the decoder $D_w$ to make the diffusion model's generated images non-watermarked using an attacking dataset of size $n$, as illustrated in Figure 3. In Step I, we estimate the denoised latent vector $z^i$ for each non-watermarked image $x^i$ in the attacking dataset, where $i = 1, 2, \ldots, n$. In Step II, by utilizing these images and their estimated denoised latent vectors, we fine-tune the decoder $D_w$ to ensure that the reconstructed images closely match the non-watermarked images when the inputs are the corresponding estimated denoised latent vectors. Our intuition is that a watermarked decoder will transform a denoised latent vector $z^i$ to the watermarked version of $x^i$, denoted as $x_w^i$. Therefore, through fine-tuning the decoder to reconstruct $x^i$ from the input $z^i$, the decoder is trained to map any given denoised latent vector to the non-watermarked version of its corresponding image, effectively removing watermarks from images generated by the diffusion model.

### 4.2 STEP I: ESTIMATE THE DENOISED LATENT VECTOR $z$

To estimate the denoised latent vector $z^i$ for the non-watermarked image $x^i$, we propose different methods in different scenarios.

**E-aware:** In this scenario, an attacker has access to the encoder $E$, diffusion process $DP$, denoising layers $DN$, and watermarked decoder $D_w$. Based on the pipeline of the diffusion model, the denoised latent vector $z^i$ can be represented as $z^i = DN(DP(E(x^i)), c^i)$. However, since we don't have access to the ground-truth conditioning $c^i$ to reconstruct $z^i$, we cannot directly compute $z^i$ even though we have access to $E$, $DP$, and $DN$. We observe that the denoising layers $DN$ are trained to denoise the noisy latent vector $z_T$ such that $DN(z_T, c)$ is close to $E(x)$. Therefore, the attacker can utilize the encoder to encode the non-watermarked image $x^i$ into the latent space to get

an estimation of the denoised latent vector $z^i$, denoted by $\hat{z}^i$, as follows:

$$\hat{z}^i = E(x^i), \forall i. \tag{1}$$

**E-agnostic:** In this scenario, an attacker only has access to the denoising layers $DN$ and watermarked decoder $D_w$. The most straightforward way to estimate the denoised latent vector $z^i$ is to train a new encoder based on $DN$ and $D_w$ and use the method in E-aware scenario. However, training an encoder from scratch for a latent diffusion model to achieve good encoding performance requires a large number of data and computational resources, which is very time-consuming and infeasible for an attacker with limited resources. Recall that our goal is to estimate the denoised latent vector $z^i$ which will be mapped to the watermarked image $x_w^i$ by the watermarked decoder $D_w$. Formally, we can formulate an equation as follows:

$$D_w(z^i) = x_w^i, \forall i. \tag{2}$$

This equation is difficult to solve since there are two variables in it, the denoised latent vector $z^i$ and watermarked image $x_w^i$. To reduce the number of variables, we use the known $x^i$ as an approximation of $x_w^i$ since the watermarked version of an image should be highly perceptually close to the non-watermarked version. Therefore, to get an estimation of $z^i$, we can reformulate the equation as follows:

$$D_w(\hat{z}^i) = x^i, \forall i. \tag{3}$$

We can easily get an estimation of $z^i$ for Equation 3 if $D_w$ is invertible, i.e., $\hat{z}^i = D_w^{-1}(x^i), \forall i$. However, since the diffusion model's decoder is a complicated neural network and it is usually infeasible to get its inverse function, solving the Equation 3 directly is challenging. To address the challenge, we can treat $\hat{z}^i$ as a trainable variable and reformulate Equation 3 into an optimization problem as follows:

$$\min_{\hat{z}^i} l_p(D_w(\hat{z}^i), x^i), \forall i, \tag{4}$$

where $l_p(\cdot, \cdot)$ denotes the perceptual loss between two images to ensure the visual similarity. However, it is still challenging to make $D_w(\hat{z}^i)$ closely resemble the non-watermarked image $x^i$ since $\hat{z}^i$ is randomly initialized and $D_w(\hat{z}^i)$ is completely different from $x^i$ at the early stage of the optimization process.

Therefore, we propose a two-stage optimization method to solve the optimization problem described in Equation 4. At the first stage, for each $\hat{z}^i$, we randomly initialize it using a standard Gaussian distribution. Then we employ gradient descent to find an initial point $\hat{z}_{init}^i$ for $\hat{z}^i$ that minimizes the mean square error between $D_w(\hat{z}_{init}^i)$ and $x^i$. This stage ensures that $D_w(\hat{z}_{init}^i)$ roughly resembles $x^i$, though with a significant loss of detailed information. At the second stage, we initialize $\hat{z}^i$ with the initial point $\hat{z}_{init}^i$ obtained from the first stage. Then we set $l_p(\cdot, \cdot)$ to be the Watson-VGG perceptual loss (Czolbe et al., 2020) and use gradient descent to further optimize $\hat{z}^i$, enabling it to capture the detailed information of the non-watermarked image $x^i$. The detailed method to estimate the denoised latent vector $z^i$ in E-agnostic scenario is shown in Algorithm 1 in Appendix.

### 4.3 STEP II: FINE-TUNE THE DECODER $D_w$

Given a set of estimated denoised latent vectors $\hat{z}^i$ and non-watermarked images $x^i$, our goal is to modify the parameters of the watermarked decoder $D_w$ to make the diffusion model's generated images non-watermarked. The main idea is to modify the decoder's parameters to enable it to map the denoised latent vector $z^i$, which is originally mapped to the watermarked image $x_w^i$, to the non-watermarked image $x^i$. To achieve this, we use the estimated denoised latent vectors $\hat{z}^i$ and non-watermarked images $x^i$ to fine-tune the decoder, ensuring that the reconstructed images closely resemble the non-watermarked images at the pixel level to effectively remove the watermark signal from each pixel. Formally, we can formulate the optimization problem as follows:

$$\min_{D_w} \frac{1}{n} \sum_{i=1}^{n} \|D_w(\hat{z}^i) - x^i\|_2. \tag{5}$$

However, since the mean square error measures the average difference between the non-watermarked and reconstructed images, it tends to penalize large errors more severely than small ones, leading to a smoothing effect where the reconstructed images may lose lots of detailed information. To solve this challenge, a perceptual loss that measures the distance of the high-level features produced by a pre-trained neural network between two images is employed to ensure the visual quality of the reconstructed images. Formally, we can reformulate the optimization problem as follows:

$$\min_{D_w} \frac{1}{n} \sum_{i=1}^{n} \|D_w(\hat{z}^i) - x^i\|_2 + \lambda \frac{1}{n} \sum_{i=1}^{n} l_p(D_w(\hat{z}^i), x^i), \tag{6}$$

where $\lambda$ denotes the weight for the perceptual loss. To solve the optimization problem, we employ gradient descent to optimize the parameters of $D_w$ to minimize the objective function in Equation 6. During the optimization, we adopt a convolution neural network introduced by Zhu et al. (2018) as a discriminator to perform adversarial training. The discriminator is trained to distinguish $D_w(\hat{z}^i)$ from $x^i$ and the decoder $D_w$ is trained to fool the discriminator. Formally, we reformulate the optimization problem as follows:

$$\min_{D_w} \frac{1}{n} \sum_{i=1}^{n} \|D_w(\hat{z}^i) - x^i\|_2 + \lambda \frac{1}{n} \sum_{i=1}^{n} l_p(D_w(\hat{z}^i), x^i)$$
$$+ \mu \frac{1}{n} \sum_{i=1}^{n} log(1 - disc(D_w(\hat{z}^i))), \tag{7}$$

where $disc$ denotes the discriminator and $\mu$ denotes the weight for the adversarial loss. The detailed method to fine-tune the decoder $D_w$ is shown in Algorithm 2 in Appendix.

## 5 EVALUATION

### 5.1 EXPERIMENTAL SETUP

**Datasets:** We employ public non-AI-generated images as our attacking datasets. Specifically, we utilize three datasets: ImageNet (Russakovsky et al., 2015), MS-COCO (Lin et al., 2014), and Conceptual Captions (Sharma et al., 2018). From each dataset, we randomly select 4,000 images as an attacking dataset to fine-tune the watermarked decoder. The images in the attacking datasets are resized to $256 \times 256$. For testing, we evaluate the effectiveness and utility goals using images generated by an open-source watermarked diffusion model and its versions fine-tuned by watermark removal attacks. These images are produced using text prompts from the Stable Diffusion Prompts dataset created by MagicPrompt (Santana, 2023). Specifically, we randomly sample 1,000 text prompts from the dataset to generate 1,000 images for testing.

**Detecting watermark in an image:** In our experiments, we consider *double-tail detector* (Jiang et al., 2023), which is a more robust version of watermark-based detector, as introduced in detail in Appendix A.1.

**Diffusion model and watermarking decoder:** We evaluate two recent watermarking methods designed for open-source diffusion models: Stable Signature (Fernandez et al., 2023) and WOUAF (Kim et al., 2024). For Stable Signature, we use the open-source Stable Diffusion 2.1 model and its watermarked version produced by Stable Signature. For WOUAF, we use the open-source Stable Diffusion 2-base model and its watermarked version produced by WOUAF's mapping network (Kim et al., 2024). Further details on both methods are provided in Appendix A.2. For the watermarking decoder $W_d$, we use the respective open-source decoders provided by Stable Signature and WOUAF. Unless otherwise mentioned, we adopt Stable Signature as the default watermarking method.

**Different variants to estimate the denoised latent vector $z$:** In our experiments, we compare our two-stage optimization method (denoted by 2S) with the variants shown in Appendix A.3 to estimate the denoised latent vector $z$.

**Per-image-based removal attacks:** In our experiments, we compare our attack with five commonly used per-image-based removal attacks, including the state-of-the-art one proposed by Jiang et al.

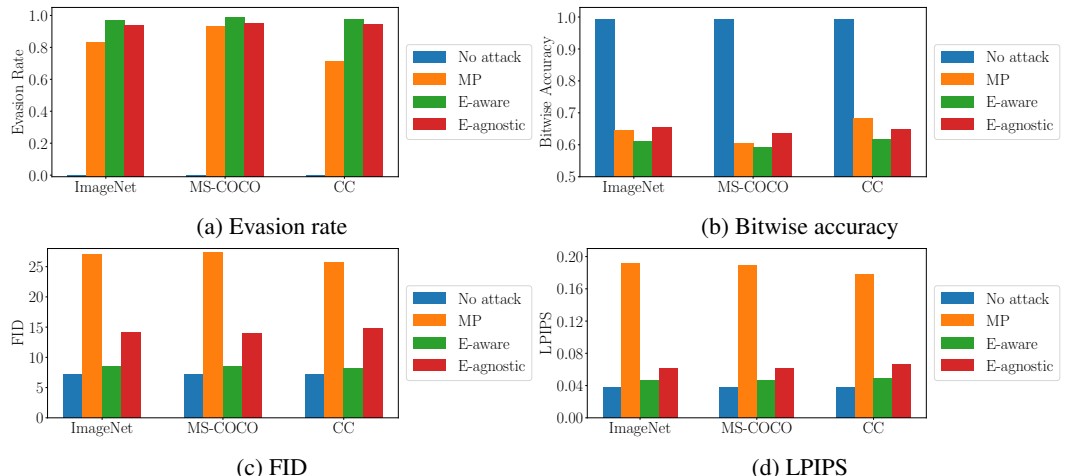

Figure 4: Effectiveness and utility of MP and our attack on Stable Signature with the three attacking datasets.

(2023). The details of the per-image-based removal attacks we use are shown in Appendix A.4. It should be emphasized that all of these per-image-based attacks require to craft a perturbation for each watermarked image individually to remove watermark.

**Model-targeted removal attack:** For model-targeted attacks, we compare our attack with MP, the only existing model-targeted attack, introduced in Stable Signature (Fernandez et al., 2023). Note that this method requires the access to the diffusion model's encoder and is only applicable in the E-aware scenario, which is introduced in detail in Appendix A.5.

**Evaluation metrics:** To evaluate whether our attack achieves the effectiveness goal, we utilize two metrics: *evasion rate* and *bitwise accuracy*. Additionally, to evaluate whether our attack achieves the utility goal, we use two commonly used metrics for the generation quality of generative models, i.e., *Fréchet Inception Distance (FID)* and *LPIPS* (Zhang et al., 2018). The details of the evaluation metrics are shown in Appendix A.6.

**Parameter settings:** In our experiments, 2S is employed as the default method to estimate the denoised latent vector $z$ in the E-agnostic scenario. Given that the watermark length in our experiments is 48, $\tau$ is set to be 0.77 to ensure that the false positive rate of the double-tail detector does not exceed $10^{-4}$. The detailed parameter settings for our experiments are shown in Appendix A.7.

## 5.2 EXPERIMENTAL RESULTS

**Our attack achieves both the effectiveness and utility goals:** Figures 4 and 5 show the evasion rate, bitwise accuracy, FID, and LPIPS for MP and our attack across the three attacking datasets on Stable Signature and WOUAF, respectively. First, we observe that our attack effectively evades watermark-based detection in both E-aware and E-agnostic scenarios. For Stable Signature, the evasion rate exceeds 94%, with a bitwise accuracy below 66%, while maintaining an FID lower than 14.79 and an LPIPS under 0.066. Similarly, for WOUAF, the evasion rate reaches 100%, with a bitwise accuracy below 57%, while maintaining an FID below 18.1 and an LPIPS under 0.077. Notably, in the E-aware scenario, our attack produces images with lower FID and LPIPS than the watermarked images produced by WOUAF without attack. This improvement occurs because WOUAF compromises the original image quality when embedding the watermark. Our attack recovers these images from the degradation, thereby enhancing their quality.

Second, we observe that our attack outperforms MP in both scenarios. In the E-aware scenario, our attack achieves a higher evasion rate and lower bitwise accuracy, while consistently maintaining a significantly lower FID and LPIPS across all three attacking datasets. In the E-agnostic scenario, our attack still achieves a comparable or higher evasion rate and comparable bitwise accuracy, while continuing to maintain a much lower FID and LPIPS in all datasets. It is important to note that MP

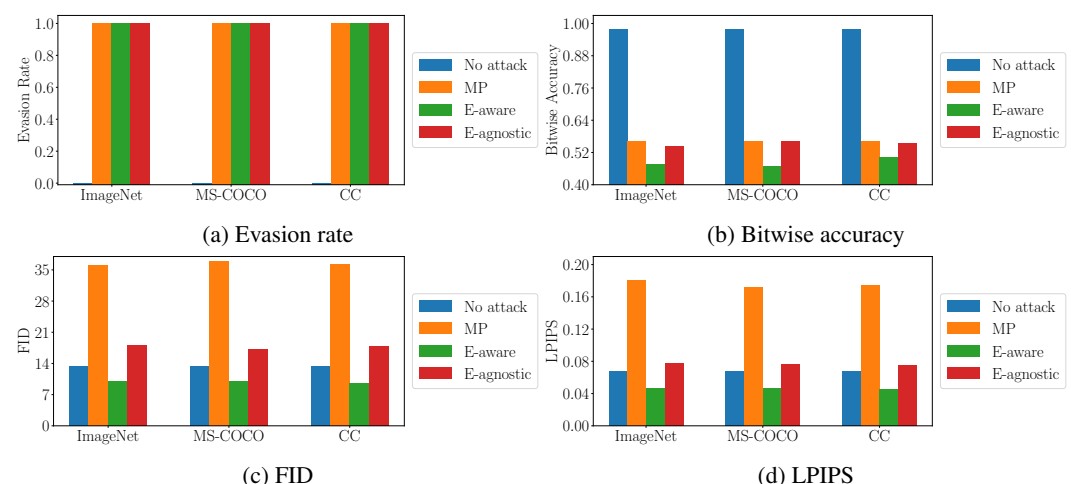

Figure 5: Effectiveness and utility of MP and our attack on WOUAF with the three attacking datasets.

Table 1: Utility and processing time of per-image-based attacks and our attack.

| Method | Utility | | | Time | |
|---|---|---|---|---|---|
| | FID ↓ | LPIPS ↓ | PSNR ↑ | Fine-tuning (min) ↓ | Removal (s/img) ↓ |
| JPEG | 31.79 | 0.283 | 27.28 | - | 0.036 |
| Brightness | 112.84 | 0.688 | 5.25 | - | 0.005 |
| Contrast | 88.25 | 0.557 | 10.03 | - | 0.002 |
| GN | 132.92 | 1.145 | 12.99 | - | 0.017 |
| WEvade-W-II | 8.66 | 0.051 | 29.56 | - | 651.034 |
| E-aware | 8.48 | 0.047 | 29.51 | 14.197 | - |
| E-agnostic | 14.15 | 0.061 | 28.76 | 8777.885 | - |

assumes the attacker has access to the encoder, whereas our attack in the E-agnostic scenario does not. Figure 8 and 9 in Appendix provide image examples comparing our attack with the clean and watermarked images by Stable Signature and WOUAF. We observe that the images produced by our non-watermarked decoder are nearly indistinguishable from those generated by the clean and watermarked decoders.

**Comparing with per-image-based removal attacks:** Table 1 shows the utility and the processing time of our attack compared with five per-image-based removal attacks when achieving similar evasion rate and bitwise accuracy. Figure 10 in Appendix shows the generated (or perturbed) images by different attacks. We also show the *Peak signal-to-noise ratio (PSNR)*, a common metric for assessing per-image-based attacks' utility. The processing time is divided into decoder fine-tuning and watermark removal phases. Note that our attack's fine-tuning time, measured on a single NVIDIA A6000 GPU, can be significantly reduced using multiple GPUs. For instance, with four NVIDIA A6000 GPUs, fine-tuning in the E-agnostic scenario takes about 2K minutes.

First, our attack's utility surpasses most per-image-based removal attacks. Second, the removal time is 0 once the decoder is fine-tuned, making our method highly efficient for large numbers of generated images. For instance, our attack outperforms WEvade-W-II when processing more than one image in the E-aware scenario and 809 images in the E-agnostic scenario. Note that WEvade-W-II requires the access to the watermarking decoder $W_d$ to perform a white-box attack, and it represents the upper bound of the utility that can be achieved by a removal attack. Our attack achieves similar utility to WEvade-W-II when compared to clean, non-watermarked images, as we optimize the decoder's output to be closer to the non-watermarked image. It is difficult for human's eyes to notice their differences, as shown in Figure 10 in Appendix.

**Different variants to estimate $z$:** Figure 6 shows the FID and examples of reconstructed images by different methods to estimate the denoised latent vector $z$ for non-watermarked images. The FID is

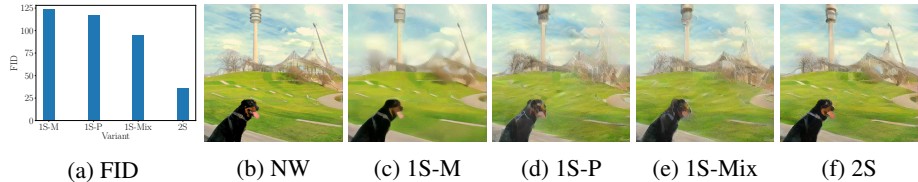

(a) FID     (b) NW     (c) 1S-M     (d) 1S-P     (e) 1S-Mix     (f) 2S

Figure 6: Image reconstruction performance for different variants to estimate $z$ on ImageNet. NW denotes the non-watermarked image.

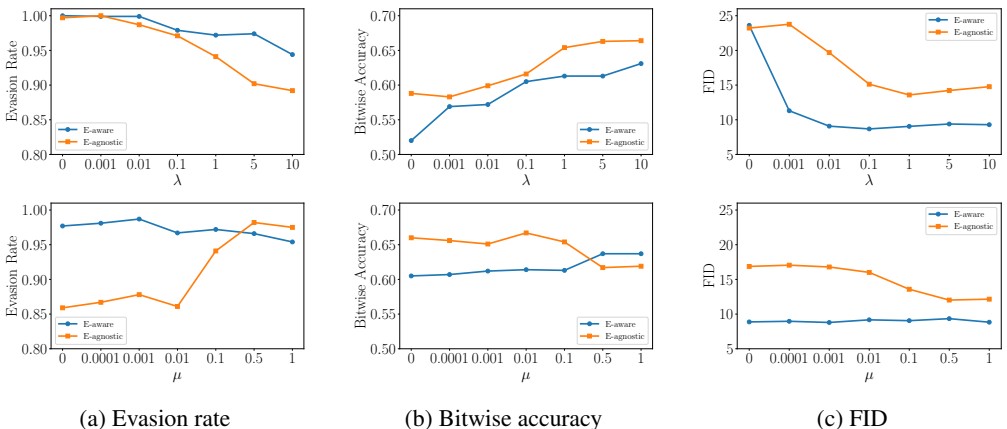

(a) Evasion rate        (b) Bitwise accuracy        (c) FID

Figure 7: Effectiveness and utility of our attack with different $\lambda$ (first row) and $\mu$ (second row) values on ImageNet.

calculated between 100 randomly selected ImageNet images and their reconstructed versions. The 2S method produces images more similar to the originals and achieves a much lower FID than other methods. The examples also show that $z$ from our method retains more detail and achieves higher visual similarity to original images.

**Different $\lambda$:** The first row of Figure 7 shows the evasion rate, bitwise accuracy, and FID for different $\lambda$ values in our attack. We observe that increasing $\lambda$ reduces the effectiveness of the attack because the loss function emphasizes perceptual loss over mean square error, hindering watermark removal. Initially, utility improves with larger $\lambda$ as the weight on perceptual loss increases. However, further increases in $\lambda$ lead to worse utility since focusing more on perceptual loss causes the reconstructed image to deviate pixel-wise from the non-watermarked image.

**Different $\mu$:** The second row of Figure 7 shows the evasion rate, bitwise accuracy, and FID for different $\mu$ values in our attack. In the E-aware scenario, effectiveness remains constant initially and then decreases, while utility does not change as $\mu$ increases. This occurs because small $\mu$ values already make the reconstructed image similar to the non-watermarked one, so further increases in $\mu$ do not provide additional benefits. Larger $\mu$ values also reduce the mean square error's ability to remove watermarks, decreasing effectiveness. In the E-agnostic scenario, both effectiveness and utility initially remain unchanged but later improve with larger $\mu$, as the initial reconstructed image is significantly different from the non-watermarked one, and larger $\mu$ values make them more similar.

## 6   CONCLUSION AND FUTURE WORK

In this work, we find that image watermark for open-source diffusion model is not robust as previously thought. Given a watermarked diffusion model, an attacker can remove the watermark from it by strategically fine-tuning its decoder. Our results show that our attack achieves both the effectiveness and utility goals in removing watermark from diffusion models in both E-aware and E-agnostic scenarios, and outperforms the existing model-targeted attack which is only applicable to E-aware scenario. Interesting future work is to design a more robust image watermarking method for open-source diffusion models.

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

---

**Algorithm 1** Estimate the denoised latent vector $z$

---

**Input:** Non-watermarked images $\{x^i\}_{i=1}^n$, watermarked decoder $D_w$, number of iteration for the first stage $n\_iter_1$, number of iteration for the second stage $n\_iter_2$, learning rate $\alpha$, perceptual loss function $l_p$
**Output:** Estimated denoised latent vectors $\{\hat{z}^i\}_{i=1}^n$
1: $Q \leftarrow \emptyset$
2: **for** $i = 1$ to $n$ **do**
3:    $\hat{z}^i \sim \mathcal{N}(0, 1)$
4:    **for** $j = 1$ to $n\_iter_1$ **do**
5:       $g \leftarrow \nabla_{\hat{z}^i} \|D_w(\hat{z}^i) - x^i\|_2$
6:       $\hat{z}^i \leftarrow \hat{z}^i - \alpha \cdot g$
7:    **for** $j = 1$ to $n\_iter_2$ **do**
8:       $g \leftarrow \nabla_{\hat{z}^i} l_p(D_w(\hat{z}^i), x^i)$
9:       $\hat{z}^i \leftarrow \hat{z}^i - \alpha \cdot g$
10:   $Q \leftarrow Q \cup \{\hat{z}^i\}$
11: return $Q$

---

---

**Algorithm 2** Fine-tune the decoder $D_w$

---

**Input:** Non-watermarked images $\{x^i\}_{i=1}^n$, estimated denoised latent vectors $\{\hat{z}^i\}_{i=1}^n$, watermarked decoder $D_w$, number of epoch $n\_epoch$, decoder learning rate $\alpha$, discriminator learning rate $\beta$, perceptual loss function $l_p$, discriminator $disc$, weight for perceptual loss $\lambda$, weight for adversarial loss $\mu$
**Output:** Non-watermarked decoder $D_{nw}$
1: $D_{nw} \leftarrow D_w$
2: **for** $i = 1$ to $n\_epoch$ **do**
3:    $g_{disc} \leftarrow -\nabla_{disc} \frac{1}{n} \sum_{i=1}^n [log(1 - disc(D_{nw}(\hat{z}^i))) + log(disc(x^i))]$
4:    $disc \leftarrow disc - \beta \cdot g_{disc}$
5:    $g \leftarrow \nabla_{D_{nw}} \frac{1}{n} \sum_{i=1}^n \|D_{nw}(\hat{z}^i) - x^i\|_2 + \lambda \frac{1}{n} \sum_{i=1}^n l_p(D_{nw}(\hat{z}^i), x^i) + \mu \frac{1}{n} \sum_{i=1}^n log(1 - disc(D_{nw}(\hat{z}^i)))$
6:    $D_{nw} \leftarrow D_{nw} - \alpha \cdot g$
7: return $D_{nw}$

---

# A  DETAILS OF EVALUATION

## A.1  DETAILS OF DETECTING WATERMARK IN AN IMAGE

A watermarking decoder $W_d$ is used to detect whether $w_g$ is in an image $x$. Specifically, $W_d$ is used to decode a watermark, represented as $W_d(x)$, from the image $x$. The bitwise accuracy $BA(w_1, w_2)$ between two watermarks $w_1$ and $w_2$ is the proportion of bits that are identical in $w_1$ and $w_2$. $x$ is detected as watermarked with $w_g$ if the bitwise accuracy $BA(W_d(x), w_g)$ exceeds a detection threshold $\tau$ or falls below $1 - \tau$, i.e., $BA(W_d(x), w_g) > \tau$ or $BA(W_d(x), w_g) < 1 - \tau$. Such detector is known as double-tail detector (Jiang et al., 2023), which is more robust than *single-tail detector* that detects the image $x$ as watermarked if the bitwise accuracy $BA(W_d(x), w_g)$ exceeds $\tau$. Therefore, we use double-tail detector in this work.

## A.2  DETAILS OF THE WATERMARKED DIFFUSION MODELS

For the watermarked version of Stable Diffusion 2.1 obtained through Stable Signature, the watermarked decoder $D_w$ is fine-tuned from the clean decoder $D_c$ of Stable Diffusion 2.1 using the MS-COCO dataset. The images generated by this watermarked model are embedded with a 48-bit ground-truth watermark $w_g$. For the watermarked Stable Diffusion 2-base produced by WOUAF, the watermarked decoder $D_w$ is generated through WOUAF's mapping network and weight modulation (Karras et al., 2020; Yu et al., 2020). The mapping network converts the watermark into a latent embedding, and WOUAF applies weight modulation to the clean decoder $D_c$, transforming

Figure 8: Image generated by the clean Stable Diffusion 2.1 (first row), Stable Diffusion 2.1 watermarked by Stable Signature (second row), watermarked Stable Diffusion 2.1 fine-tuned by our attack in E-aware scenario (third row), and watermarked Stable Diffusion 2.1 fine-tuned by our attack in E-agnostic scenario (fourth row). The same denoised latent vector is used by all diffusion models' decoders to generate the images in the same column. The watermark can only be detected in the images generated by Stable Diffusion 2.1 watermarked by Stable Signature (second row).

it into the watermarked decoder $D_w$. The images generated by this watermarked model contain a 32-bit ground-truth watermark $w_g$.

### A.3 OTHER VARIANTS TO ESTIMATE THE DENOISED LATENT VECTOR $z$

In our experiments, we compared our 2S method with the following variants. All of these methods initialize $\hat{z}$ with a standard Gaussian distribution and treat it as a trainable variable.

- **One-stage mean square error (1S-M)** This method optimizes $\hat{z}$ to minimize the mean square error between the reconstructed image $D_w(\hat{z})$ and the non-watermarked image $x$.
- **One-stage perceptual loss (1S-P)** This method optimizes $\hat{z}$ to minimize the perceptual loss calculated by the Watson-VGG model between $D_w(\hat{z})$ and $x$.
- **One-stage mixed loss (1S-Mix)** This method optimizes $\hat{z}$ to minimize the mixed loss consisting of mean square error and perceptual loss calculated by the Watson-VGG model between $D_w(\hat{z})$ and $x$. The weights for different loss functions are set to be 1.

### A.4 DETAILS OF PER-IMAGE-BASED REMOVAL ATTACKS

- **JPEG** It is a commonly used image compression technique that can significantly decrease the size of image files while preserving high image quality. The quality of images processed by JPEG is governed by a quality factor. Using a smaller quality factor to post-process watermarked images can make the detection of watermarks within the image more difficult.

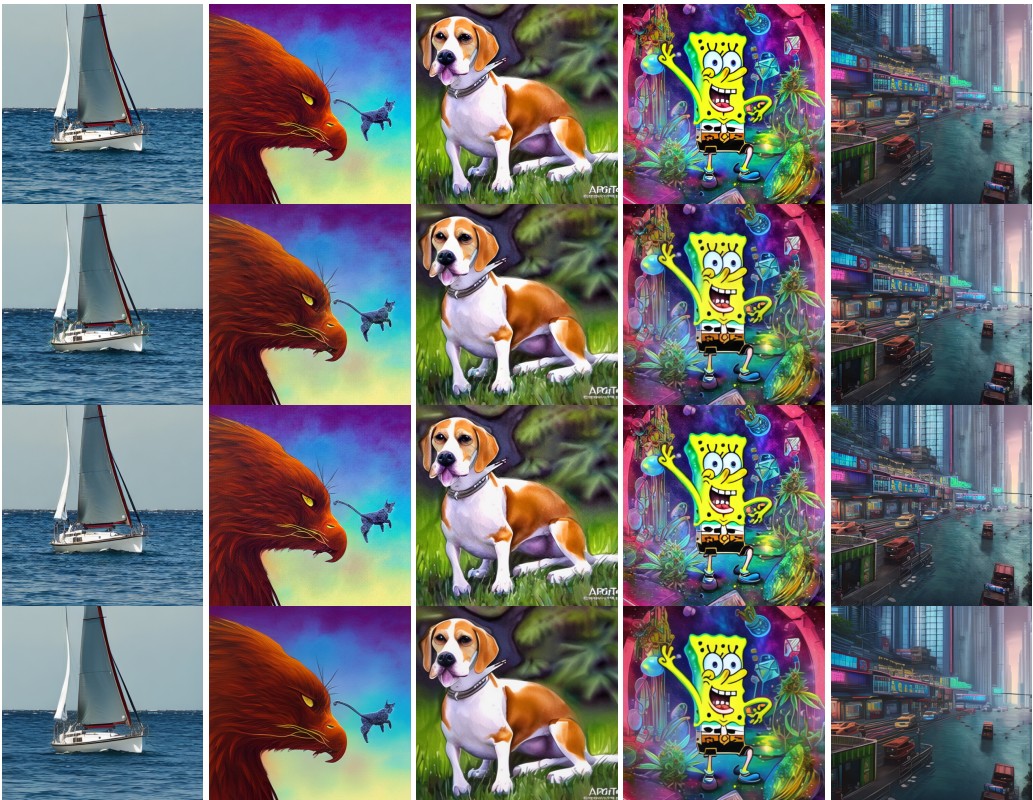

Figure 9: Image generated by the clean Stable Diffusion 2-base (first row), Stable Diffusion 2-base watermarked by WOUAF (second row), watermarked Stable Diffusion 2-base fine-tuned by our attack in E-aware scenario (third row), and watermarked Stable Diffusion 2-base fine-tuned by our attack in E-agnostic scenario (fourth row). The same denoised latent vector is used by all diffusion models' decoders to generate the images in the same column. The watermark can only be detected in the images generated by Stable Diffusion 2-base watermarked by WOUAF (second row).

- **Brightness** This method modifies the brightness of an image by initially converting the image to a color space that includes a brightness-related channel. It then isolates this channel, adjusts its intensity by multiplying it with a specified factor, and finally converts the image back to its original color space. This method may disrupt the watermark patterns in watermarked images to evade watermark detection.

- **Contrast** This method alters the contrast of an image by modifying its pixel values. Specifically, for each pixel, it subtracts 127 from the pixel's value, multiplies the result by a factor $k$, and then adds 127 to the outcome. The factor $k$ determines the level of contrast enhancement or reduction, with values greater than 1 increasing contrast and values between 0 and 1 decreasing it.

- **Gaussian noise (GN)** This method adds a noise that follows a Gaussian distribution with a zero mean and a standard deviation of $\sigma$ to the watermarked image. It simulates the noise effects commonly encountered in the real world. A larger $\sigma$ value makes it more challenging to detect watermarks, simultaneously compromising image quality.

- **WEvade-W-II (Jiang et al., 2023)** This method employs projected gradient descent (PGD) to optimize a perturbation applied to the watermarked image such that the decoded watermark from the perturbed image by the model provider's watermarking decoder closely matches a randomly generated watermark, with each bit uniformly sampled from $\{0, 1\}$. We assume that the attacker has access to the watermarking decoder for this method.

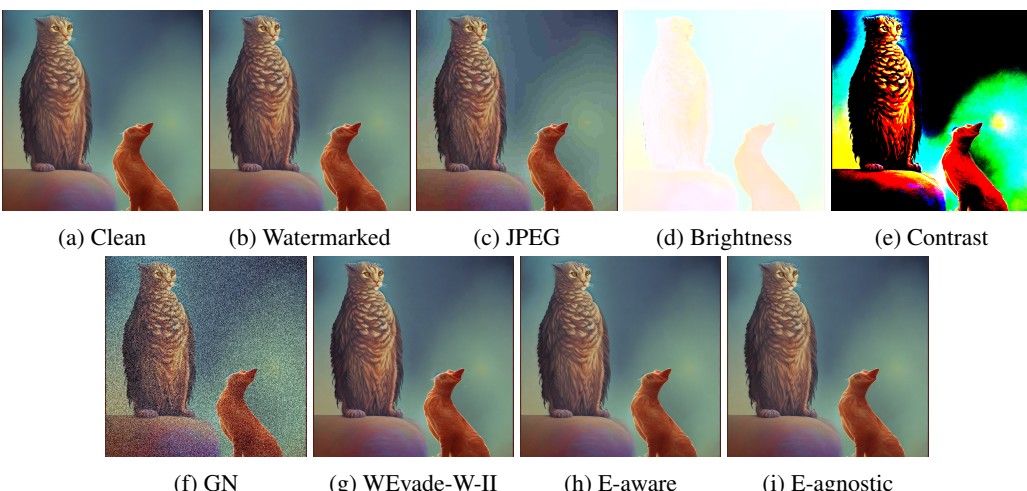

| (a) Clean | (b) Watermarked | (c) JPEG | (d) Brightness | (e) Contrast |

| (f) GN | (g) WEvade-W-II | (h) E-aware | (i) E-agnostic |

Figure 10: An example of generated image (a) with clean decoder, (b) with watermarked decoder, (c) with watermarked decoder attacked by JPEG, (d) with watermarked decoder attacked by Brightness, (e) with watermarked decoder attacked by Contrast, (f) with watermarked decoder attacked by GN, (g) with watermarked decoder attacked by WEvade-W-II, (h) with non-watermarked decoder fine-tune by our attack in E-aware scenario, (i) with non-watermarked decoder fine-tune by our attack in E-agnostic scenario. The watermark can only be detected in (b).

## A.5 DETAILS OF MP

MP involves fine-tuning the diffusion model's encoder and decoder with the encoder's parameters fixed to reconstruct non-watermarked images using mean square error as the reconstruction loss. Following the configuration by Fernandez et al. (2023), we employ AdamW and a learning rate of 0.0005 with a linear warm-up period of 20 iterations followed by a half-cycle cosine decay to fine-tune the decoder with a batch size of 4 to achieve similar bitwise accuracy on the attacking dataset as our attack in the E-aware scenario.

## A.6 DETAILS OF EVALUATION METRICS

The evasion rate refers to the proportion of generated images (or perturbed images, in the case of per-image-based removal attacks) that are detected as watermarked by the watermark-based detector. Bitwise accuracy measures the proportion of bits in the watermark decoded from a generated (or perturbed) image that match the ground-truth watermark $w_g$. For the FID score, we calculate it on the test set by comparing the generated (or perturbed) images to the original images produced by the clean Stable Diffusion 2.1 model using the same random seed. Similarly, LPIPS is computed by comparing the generated (or perturbed) images to the original images generated by the clean Stable Diffusion 2.1, also using the same random seed. Both bitwise accuracy and LPIPS are averaged across 1,000 images in the test set.

## A.7 DETAILS OF PARAMETER SETTINGS

In the E-aware scenario, we use the Watson-VGG (Czolbe et al., 2020) model to measure the perceptual loss in Step II. However, in Step I of our attack, we use the Watson-VGG model to measure the perceptual loss in the E-agnostic scenario. To avoid potential local minima issues that could emerge from using the same perceptual loss model, we use VGG-16 (Falbel, 2024) to measure the perceptual loss in E-agnostic scenario in Step II. For the discriminator $disc$, we employ the discriminator in HiDDeN (Zhu et al., 2018).

To estimate the denoised latent vector $z$ in the E-agnostic scenario, we execute 500 epochs for each stage of 2S. In each stage, the Adam optimizer, with a learning rate of 0.1, is used to optimize $\hat{z}$. For other variants to estimate $z$, we execute 1,000 epochs—equivalent to the total epoch count in 2S—and maintain consistent optimizer settings.

For decoder fine-tuning, we execute 1 epoch in the E-aware scenario and 2 epochs in the E-agnostic scenario. We set the parameters $\lambda = 1$ and $\mu = 0.1$. Additionally, the AdamW optimizer is used, with a base learning rate of 0.0005 with a linear warm-up period of 20 iterations followed by a half-cycle cosine decay. The batch size is set to be 4. For optimizing the discriminator, the Adam optimizer is used with a learning rate of 0.001.

