# OpenReview forum: "Stable Signature is Unstable: Removing Image Watermark from Diffusion Models"
_ICLR.cc/2025/Conference — ICLR 2025 Conference Withdrawn Submission_

### Official Review · Reviewer_CJQL · 2024-11-01

**Soundness:** 2
**Presentation:** 3
**Contribution:** 2
**Rating:** 5
**Confidence:** 4

**Summary:**

This paper presents a watermark removal attack targeting images generated by open-source diffusion models. The authors identify the limitations of existing in-generation watermark methods, such as Stable Signature, in the context of removal attacks. They explore two scenarios regarding the attacker's access to the encoder \( E \) and propose a novel two-step attack method. Experimental results demonstrate that their approach can effectively remove watermarks from diffusion model-generated images while maintaining visual quality.

**Strengths:**

1. The authors clearly articulate their motivation, and the introduction and related work sections are well-structured, allowing readers unfamiliar with the field to quickly grasp the background. The problem formulation is precise, with explicit statements regarding the attacker's goals and capabilities.
2. The experiments validate the effectiveness of the authors' E-aware and E-agnostic methods, particularly the E-aware attack on WOUAF, which achieves better FID and LPIPS scores than the "No attack" baseline.

**Weaknesses:**

1. The proposed method has limitations in practical applications. Both the E-aware and E-agnostic approaches require the involvement of a watermarked decoder \( D_w \), rendering them unsuitable for commercial closed-source watermark models.
2. The comparative experimental setup is insufficient, especially regarding comparisons with per-image-based removal attacks. Although the authors cite several recent per-image-based attacks in Section 2.3, they only select WEvade for comparison, neglecting other diffusion-based watermark removal methods or optimization-based approaches. A more comprehensive and extensive comparison in the experimental section should be conducted.

**Questions:**

The authors aim to eliminate watermark information from generated images by fine-tuning the decoder. However, I have a concern regarding whether the watermark information is genuinely removed or simply hidden. Since the authors only fine-tuned the diffusion model's decoder, this fine-tuning will certainly alter the distribution of the generated images. Meanwhile, the watermark model's decoder (in the case of Stable Signature, this is HiDDeN's decoder) is not fine-tuned in tandem, which would naturally lead to a decrease in watermark extraction accuracy.

If the watermark model's decoder were also fine-tuned simultaneously, would the authors' attack still remain effective? If it were still effective, this would imply that the authors' attack merely changes the distribution rather than genuinely removing the watermark information. In such a case, defenders could train a watermark model decoder to counter this attack.

---

### Official Review · Reviewer_EQbM · 2024-11-03

**Soundness:** 4
**Presentation:** 4
**Contribution:** 3
**Rating:** 5
**Confidence:** 4

**Summary:**

This paper proposes an novel method to remove watermarks in generated images of diffusion model. Experiments show that it can remove watermarks as the paper states.

**Strengths:**

1.This paper propose a model-targeted attack to remove watermark.
2.The method is effective at removing watermark on a large scale.
3.This method can preserve the quality of image.

**Weaknesses:**

This method is model-targeted, so the watermark removal relies on fine-tuning model. It is a white-box method.

**Questions:**

1.As stated in the weakness, the method relies on the model. The attack need to know and replace the decoder of the employed model to attack the targeted model. But it seems impossible. It seems to need two conditions. The first is that you can get the model of employed model. The second is that you can replace the decoder. Maybe it is useful when you buy a watermarked generated model from the owner and you want to escape the watermark generation. But it still need the owner give you the all model instead of an api.
The author may can make some explanation about the limitation.
2.The method seems to design for fine-tuned watermarking method such as the stable signature[1] as the author states in the title. So how this method perform to other watermarking method for diffusion models such as Tree-ring[2]. So i wonder know whether this method is only suitable for watermarking method as stable signature, or
It can still perform well in other methods.
3.There are some watermark removal for invisible watermark such as [3], what is the difference between the performance?
[1]Fernandez P, Couairon G, Jégou H, et al. The stable signature: Rooting watermarks in latent diffusion models[C]//Proceedings of the IEEE/CVF International Conference on Computer Vision. 2023: 22466-22477.
[2]Wen Y, Kirchenbauer J, Geiping J, et al. Tree-ring watermarks: Fingerprints for diffusion images that are invisible and robust[J]. arXiv preprint arXiv:2305.20030, 2023.
[3]Saberi M, Sadasivan V S, Rezaei K, et al. Robustness of ai-image detectors: Fundamental limits and practical attacks[J]. arXiv preprint arXiv:2310.00076, 2023.

---

### Official Review · Reviewer_dWAG · 2024-11-03

**Soundness:** 2
**Presentation:** 2
**Contribution:** 1
**Rating:** 3
**Confidence:** 4

**Summary:**

The paper evaluates the robustness of the "StableSignature" watermarking technique embedded in the parameters of diffusion model decoders, proposed by Meta. The authors present a new model-targeted attack that effectively removes these watermarks while preserving image quality. Their results reveal that the robustness of Stable Signature is overestimated, highlighting potential vulnerabilities in in-generation watermarking techniques.

**Strengths:**

1.This paper introduces a novel method of fine-tuning the decoder of a diffusion model to remove watermarks generated during the process. It demonstrates significant improvements in watermark removal while maintaining the visual quality of the output images, showing an enhancement over the Model Purification (MP) approach.

2. Comprehensive Evaluation: The study includes a thorough evaluation of the proposed attack across various scenarios (both encoder-aware and encoder-agnostic), datasets, and comparisons with existing methods like MP (Model Purification).

**Weaknesses:**

1. This paper's main finding that "Stable Signature is not robust" is not novel; in fact, previous methods[2,3] have already made the same discovery. Existing watermark removal technologies such as DiffPure[1] and Controlgen[2] can remove Stable Signature watermarks through a simple forward pass. This contrasts with the attack method proposed in this paper, which requires multiple optimization iterations and potentially access to model components like the VAE and watermark decoder.

2. The method requires fine-tuning of the model's decoder and, in some cases, access to components such as the encoder or VAE. This requirement may not always be practical for real-world adversaries and limits the broader applicability of the proposed attack. The time overhead is extremely high, making it difficult to use in practice. If I can access the decoder of a certain watermarking method, removing the watermark through optimization iterations and image regularization is an expected outcome and lacks innovation.

3. This paper can only remove a single type of watermark.

Reference:
1. Diffusion Models for Adversarial Purification
2. Image Watermarks are Removable Using Controllable Regeneration from Clean Noise
3. WAVES: Benchmarking the Robustness of Image Watermarks

**Questions:**

1. Given that some existing methods can remove watermarks with simpler forward processes, how does your approach justify its more complex optimization steps in terms of practical use cases?

2. The evaluation shows that the watermark removal performance varies under different conditions. How do you see your approach being adapted or scaled to scenarios where full access to the VAE or other internal components is not feasible?

---

### Official Review · Reviewer_8r4h · 2024-11-04

**Soundness:** 3
**Presentation:** 3
**Contribution:** 2
**Rating:** 5
**Confidence:** 3

**Summary:**

The paper introduces a method to remove watermarks added by stable signature [1]. The method achieves high removal effect and good image quality compared to previous method MP [2]. The method builds on the assumption that the attacker can modify the parameters of the open-sourced watermarked latent diffusion model decoder. The approach consists of two steps: first, the method estimates a denoised latent vector for each non-watermarked image in an attacking dataset; then, it fine-tunes the decoder so that the generated images align closely with the non-watermarked images. In the encoder-aware scenario, the attacker has access to the model’s encoder, diffusion process, and denoising layers. In the encore-agnostic scenario, the attacker only has access to the denoising layers and decoder. The experiment results show that this method achieves high watermark removal rates with preserved image quality.

[1] Fernandez, P., Couairon, G., Jégou, H., Douze, M., & Furon, T. (2023). The stable signature: Rooting watermarks in latent diffusion models. In Proceedings of the IEEE/CVF International Conference on Computer Vision (pp. 22466-22477).

**Strengths:**

1. The approach can adapt to two different access levels. It’s flexible and close to real-world settings.
2. The watermark demonstrates high quality in watermark removal tasks and preserves image quality. Although E-agnostic takes long time to fine-tune, the proposed method is model-targeted and therefore needs no additional processing time once fine-tuned.

**Weaknesses:**

1. It seems the novelty of the method is not very apparent. The method estimates the latent z and do fine-tuning on decoder based on the loss function, which is a common method in watermarking methods, see [1].
2. It’s still not very clear to me why the encoded latent $\hat{z}^i$ can be used to approximate the true $z^i$. Although the decoded image can be similar to the original image visually, the latents $\hat{z}^i$ and $z^i$ may differ significantly. suppose the distribution of the latent after the denoising process of the diffusion model is $P$, and the distribution of the latent after the encoding of a clean image using the encoder is $P'$. $z^i$ is sampled from $P$, while $\hat{z}^i$ is sampled from $P'$. It's obvious that $P$ and $P'$ is not the same, though they may be close to each other. The fine-tuning is actually a process that tries to adapt the input distribution of the decoder to $P'$.


[1] Zhang, L., Liu, X., Martin, A. V., Bearfield, C. X., Brun, Y., & Guan, H. (2024). Robust Image Watermarking using Stable Diffusion. arXiv preprint arXiv:2401.04247.

**Questions:**

1. Why the e-agnostic scenario takes so much time in fine-tuning? Can the authors specify the process?
2. I’m confused about the changing of  evasion rate and the bitwise accuracy with respect to $\mu$, the weight of the adversarial loss. It seems contradictory that as $\mu$ increases, in terms of evasion rate and bitwise accuracy, the e-aware curve and e-agnostic curve has two different trending directions. Can the authors explain it?
3. The assumption is that the attacker can access the decoder of an open-sourced model. I’m quite confused about this setting. If the decoder can be accessed, why not just replace the decoder with the original one? It seems the application scenario of this method is not necessary in real-life? Can the authors explain it?

---

### Note · Authors · 2024-11-13

I have read and agree with the venue's withdrawal policy on behalf of myself and my co-authors.